# The Use of Contrast-Enhanced Ultrasound (CEUS) in the Evaluation of the Neonatal Brain

**DOI:** 10.3390/children10081303

**Published:** 2023-07-28

**Authors:** Domen Plut, Maja Prutki, Peter Slak

**Affiliations:** 1Clinical Radiology Institute, University Medical Centre Ljubljana, 1000 Ljubljana, Slovenia; 2Department of Radiology, Faculty of Medicine, University of Ljubljana, 1000 Ljubljana, Slovenia; 3Clinical Department of Diagnostic and Interventional Radiology, University Hospital Center Zagreb, School of Medicine, University of Zagreb, 10000 Zagreb, Croatia

**Keywords:** brain, child, contrast-enhanced ultrasound (CEUS), hypoxic-ischemic encephalopathy, infection, imaging, neonate, sonography, stroke, tumor

## Abstract

In recent years, advancements in technology have allowed the use of contrast-enhanced ultrasounds (CEUS) with high-frequency transducers, which in turn, led to new possibilities in diagnosing a variety of diseases and conditions in the field of radiology, including neonatal brain imaging. CEUSs overcome some of the limitations of conventional ultrasounds (US) and Doppler USs. It allows the visualization of dynamic perfusion even in the smallest vessels in the whole brain and allows the quantitative analysis of perfusion parameters. An increasing number of articles are published on the topic of the use of CEUSs on children each year. In the area of brain imaging, the CEUS has already proven to be useful in cases with clinical indications, such as hypoxic-ischemic injuries, stroke, intracranial hemorrhages, vascular anomalies, brain tumors, and infections. We present and discuss the basic principles of the CEUS and its safety considerations, the examination protocol for imaging the neonatal brain, and current and emerging clinical applications.

## 1. Introduction

An ultrasound (US) is a type of medical imaging that is commonly used in pediatric radiology due to its multiple advantages for the imaging of young patients. Besides the traditional use of USs in the assessment of abdominal organs and other soft tissues, the open anterior fontanelle in neonates allows the evaluation of the brain. Cranial USs have become a primary screening tool for detecting intracranial pathologic conditions in infants [1]. Technical advancements, such as the use of high-frequency transducers, imaging through secondary acoustic windows, and the incorporation of Doppler imaging, have made it possible to conduct a thorough assessment of the neonatal brain, often eliminating the need for additional imaging. One disadvantage of the US is the limited ability to differentiate between soft tissues and evaluate vascularity without the use of contrast media, unlike CT and MRI. However, the development of contrast-enhanced ultrasounds (CEUS) allowed new possibilities in diagnosing a variety of diseases and conditions and may lead to even of a less need for additional imaging [2]. Doppler USs have limitations when measuring slow-flow conditions, assessing the microvasculature, and are sensitive to motion-related artifacts. The CEUS overcomes these limitations and allows the visualization of dynamic perfusion even in the smallest vessels in the whole brain. The CEUS also allows quantitative analysis to be performed [3]. It has already shown potential in applications to various clinical indications, such as hypoxic-ischemic injuries, stroke, intracranial hemorrhages, vascular anomalies, brain tumors, and infections. Future research is expected to expand its applications and facilitate its regulatory approval.

## 2. Basics of Contrast-Enhanced Ultrasound (CEUS)

The most widely utilized ultrasound contrast agent (UCA) is SonoVue^®^ (Bracco, Milan, Italy), which is marketed as Lumason^®^ in the USA. This UCA is made up of tiny bubbles (i.e., microspheres) made of sulfur hexafluoride gas enclosed by a layer of phospholipids on the outside. When they are exposed to a US pulse, these microbubbles vibrate and reflect strong signals. Imaging is carried out at low acoustic power, typically using a mechanical index (MI) below 0.1, and a high frame rate. The signals emitted by the microbubbles contain multiples of the insonating frequency or harmonics, which are used to create images of the contrast agent. If the MI is too high, it can cause the microbubbles to rupture, resulting in suboptimal enhancement. SonoVue^®^ is a blood pool agent that remains within the intravascular space. Because of the small size of the microbubbles, which are less than 6 μm, they can reach even the smallest capillaries when injected intravenously (IV). The core of the microsphere is a harmless gas that is excreted through the lungs. The phospholipid microbubble shell is broken down and excreted through the liver. The majority of the applied dose of UCA is eliminated from the body in a few minutes [4].

### 2.1. Safety Considerations

UCAs are very safe, with the frequency of adverse events lower than that of other contrast agents (i.e., CT and MRI contrast agents) [5,6]. Additionally, the rate of contrast-related adverse events in children is lower than it is in adults [7]. A recent meta-analysis of nearly five thousand IV CEUS examinations of children reported that adverse events were rare, and most of them were not severe [8]. Mild adverse events, such as a headache, nausea, altered taste, tinnitus, light headedness, urticaria, and hyperventilation, were reported in only 1.1% of the IV examinations. Severe adverse events, such as anaphylactic reactions, were extremely rare, occurring in only 0.2% of the IV examinations. Nonetheless, a contrast reaction kit should be available nearby whenever one is performing any examination with IV contrast media, including CEUS [9].

CEUSs have unique safety concerns related to bioeffects. In theory, if the MI used is too high (MI > 1.9), the microspheres may rupture during oscillation, potentially causing damage to the adjacent cells. This process is called sonoporation or microcavitation [10,11]. The oscillation of microbubbles may also cause tissue heating due to acoustic energy dissipation. However, these bioeffects occur only with high dosages of UCA, long pulse lengths, and high MIs and should not occur during diagnostic examinations due to safety precautions incorporated into the US machines designed for diagnostic clinical use [10,12].

In the USA, the IV use of UCAs by children was approved by the FDA in 2016 for the characterization of liver lesions. In Europe, the use of UCAs by children is only registered for intracavitary use in the diagnosis of vesicoureteral reflux; therefore, the IV use of UCAs is off label. However, in its 2011 recommendations, the European Federation of Societies for Ultrasound in Medicine and Biology (UFSUMB) recommended the off-label use of UCAs by children for many other indications as well, including IV use, due to their proven effectiveness and safety [13,14]. At the same time, it should be emphasized that a large proportion of medicines taken by children are used off label.

### 2.2. Contrast-Enhanced Ultrasound (CEUS) Technique

Before performing a CEUS, it is important to perform a detailed grayscale US examination to evaluate the region of concern. This initial assessment can be beneficial in customizing the subsequent CEUS protocol. Most newer US scanners have the ability to perform CEUS imaging. CEUS imaging is typically performed using a contrast-specific mode with a dual-display option. This allows the concurrent viewing of a grayscale and contrast-only image. To achieve the best detection of microbubbles, the gain setting should be barely above the noise floor, which ensures that only the microbubbles are visualized [15]. The choice of a transducer depends on the location and vascularity of the examined organ and the size of the patient. For larger and older children, convex transducers with lower frequencies are used, while neonates and infants with superficial lesions require convex transducers with higher frequencies or linear transducers. However, due to the microbubbles’ size and their oscillating frequency, using higher-frequency transducers results in a suboptimal image, which, in turn, requires a higher dose of the UCA to compensate for it [16]. 

Note that a CEUS is not only a still image technique. A CEUS is a real-time imaging technique that allows the visualization of blood flow and contrast agent distribution over time. Therefore, typically, a CEUS examination is conducted recording a cineloop. For CEUSs with the IV application of UCA, two people are typically needed, with one person acquiring the US images or cineloop, while the other prepares and administers the UCA through a central or peripheral line. The dose for pediatric IV administration approved by the FDA is 0.03 mL/kg up to 2.4 mL. The application of a UCA can be repeated once during the examination [17]. After injecting the contrast bolus, the examiner flushes the IV cannula with normal saline to push any remaining UCA through the vein and activates the timer to capture a continuous cineloop of 60–120 s. It is important to maintain the same US scan settings during image acquisition to accurately quantify the vascular perfusion parameters. In cases of a repeated examination, it is necessary to allow enough time between the injections for the UCA to clear from the body, which is usually 10–15 min [18]. The quantitative assessment of contrast kinetics can be achieved using specialized software that analyzes US images and cineloops to calculate various imaging parameters, such as the peak enhancement, time to peak enhancement, the area under the curve, regional blood volume, regional blood flow velocity, and slopes of the ascending and descending curves [19].

Brain CEUS examinations are conducted using high-frequency curved-array transducers with a frequency range of 2–11 MHz. Typically, a smaller probe that fits well within the space of the open fontanelle of an infant is used. Linear array transducers with a frequency range of 8–20 MHz are used for evaluating the extra-axial space and superficial brain structures. The image settings are optimized before contrast injection, with a low MI (<0.3) during CEUS to ensure microsphere stability. The gain is adjusted to ensure minimal background noise before contrast administration. The anterior fontanelle is typically used for still images and cineloops of the brain. A cineloop is obtained in the mid-coronal plane with the basal ganglia in view. A sweep of the entire brain is then conducted to screen the rest of the brain parenchyma. An additional microbubble injection can be used to perform a sweep through the entire brain or a specific region of interest during peak enhancement for re-evaluating or validating the findings [20,21]. For known regions of abnormality, dedicated static images and cineloops can be obtained.

### 2.3. Quantification Methods

Quantification methods can be used to measure tissue perfusions using time–intensity curve analysis. This method evaluates changes in the signal intensity over time in a region of interest, allowing for the quantification of various perfusion kinetics parameters. These parameters include the time to peak, wash-in slope, peak intensity, wash-in and washout area under the curve, and the washout slope. These parameters can be visually displayed as a color-coded map [22,23]. In a normal brain, the peak enhancement is typically achieved within 15–20 s of administering the microbubble injection, but this timing can vary depending on various factors [20]. Washout, which refers to the clearance of microbubbles from the region of interest, can occur within 10 min of administering the injection, but it may be delayed in the presence of a brain injury or other factors [24].

Another method for measuring tissue perfusion is the infusion-based destruction-replenishment method. This method involves destroying microbubbles in the field of imaging using a short acoustic pulse and studying the replenishment kinetics as circulating microbubbles flow back into the same region [3].

## 3. Clinical Applications

### 3.1. Hypoxic-Ischemic Injury

Hypoxic-ischemic encephalopathy (HIE) is a brain injury that occurs due to inadequate blood flow to an infant’s brain, which is caused by a hypoxic-ischemic event during the prenatal, intrapartum, or postnatal period. It is a serious complication that occurs in 1.5 to 2.5 out of every 1000 live births in developed countries [25]. By the age of 2, around 60% of infants with HIE may die or have severe disabilities like mental retardation, epilepsy, and cerebral palsy [26]. Despite advances in obstetric care aimed at preventing hypoxic-ischemic events, the incidence of HIE has not declined in recent years [27]. The early detection and treatment of HIE is critical for improving clinical management and predicting the short- and long-term outcomes [28]. Therapeutic hypothermia is the standard of care for this people with condition [29]. Currently, MRI is the most effective imaging technique for evaluating neurological pathology in HIE [30,31,32]. However, due to therapeutic hypothermia and technical limitations, and to achieve optimal diagnostic results, an MRI is performed several days after the onset of symptoms [33]. Studies have shown that MRIs conducted too early may underestimate the extent of the injury [34]. It is also important to note that up to 26% of neonates with HIE who underwent hypothermia and had normal MRI findings experienced abnormal neurodevelopmental outcomes [35,36]. Therefore, there is still no optimal imaging modality for the evaluation of HIE.

Brain CEUS is a promising diagnostic tool that may have advantages in comparison to MRI. Due to its bedside nature, it can be performed at any time, even during therapeutic hypothermia in an intensive care unit. Therefore, it may be used to diagnose HIE earlier and also allow monitoring the patient’s response to the treatment. The ability of CEUSs to visualize the microvasculature and evaluate brain perfusion in real time means that it has great potential to become a valuable diagnostic tool in the evaluation of HIE. There are many studies reporting the use of CEUSs for the assessment of HIE [9,24,37,38]. Recently, a quantitative CEUS approach has been introduced to screen for the presence of HIE [24]. In this study, infants with HIE were distinguished from unaffected infants by assessing the ratio of basal ganglia to cortical perfusion using wash-in, peak enhancement, and area-under-the-curve kinetic parameters on a time–intensity curve. Another recent study demonstrated that alterations in the CEUS wash-out perfusion parameters can be observed in the presence of a hypoxic-ischemic injury [38]. Future research with larger cohorts is needed to validate the diagnostic utility and prognostic value of CEUS in HIE cases. In addition, the establishment of a large CEUS database with normal brain perfusion parameters are also necessary (Figure 1).

### 3.2. Ischemic Stroke

An ischemic stroke is defined as an episode of neurological dysfunction caused by a focal cerebral infarction. An ischemic stroke among children is relatively rare, with a rate of 1.2 cases per 100,000 population per year [39]. The most affected age group are the neonates, with the reported incidence being up to six times higher [40,41]. According to population studies, the reported incidence of childhood stroke has increased in the last 20 years, most likely due to the improvements in neuroimaging techniques [42]. Non-atherosclerotic arteriopathies, prothrombotic states, and cardiac disorders, including having undergone cardiac surgery, catheterization, and extracorporeal membrane oxygenation (ECMO), account for most of the cases [43]. Childhood stroke may present with focal or diffuse signs. In the first months of life, seizures and tone abnormalities are frequent clinical hallmarks, while focal symptoms, such as numbness and weakness, or headaches are the most common clinical presentations among older children [44]. 

Advances in clinical recognition and radiological imaging methods have increased the ability to diagnose ischemic stroke among children. Imaging studies can help distinguish ischemic infarction from cerebral hemorrhage and other causes of sudden neurologic symptoms [44]. MRI has become the first-line neuroimaging modality to confirm the clinical diagnosis of ischemic stroke and should always be preferred to CT due to its greater specificity and sensitivity [40]. MRI, including diffusion-weighted imaging (DWI) and apparent diffusion coefficient (ADC) maps, can demonstrate evidence of an infarction even in the first few hours after the onset of symptoms [45]. Perfusion imaging, such as arterial spin labeling (ASL) sequence, allows the measurement of cerebral blood flow and volume and can detect areas of ischemia without the use of a contrast agent [46]. Cerebral angiography is still considered the best method to visualize the intracranial vasculature; however, continued refinements in MR angiography (MRA), especially time-of-flight angiography (TOF), which allows the visualization of flow within vessels without contrast administration, have made MRA a feasible noninvasive alternative for children, sparing the child from the potential harmful effects of ionizing radiation [47,48]. However, in critically ill neonates, the utilization of MRI can be limited due to the risks associated with transportation and incompatible support equipment. CEUSs, with their potential to evaluate brain microvasculature, present a great emerging imaging alternative. CEUSs have been used for the evaluation of acute ischemic stroke among an adult population to detect cerebral perfusion deficits and to monitor the responses to a thrombolytic treatment [49,50]. The number of studies on a pediatric population, so far, are minimal, but CEUSs have been shown to have good sensitivity to detect acute ischemic stroke in comparison to MRI [51]. For the imaging of stroke using CEUS, the quantitative brain perfusion analysis is typically performed. CEUS shows a delayed rate of the washout of the contrast agent and a delayed time-to-peak in the affected areas [49] (Figure 2). Future research via larger prospective studies is needed to better determine the diagnostic accuracy of CEUSs in comparison to MRI for the detection of stroke and its utility for the accurate characterization of the penumbra for monitoring the responses to thrombolytic therapy.

### 3.3. Intracranial Hemorrhage

An intracranial hemorrhage (ICH) in the neonatal period is a serious clinical problem and an important cause of morbidity and mortality [52]. Many inherited and acquired disorders may cause a neonatal ICH. However, in a large proportion of cases, the etiology cannot be identified [53]. Full-term neonates with an ICH commonly present with clinical features, such as apnea, bradycardia, and seizures [54,55,56]. With improvements in diagnostic imaging in recent years, even a small ICH is being increasingly recognized. The true incidence of ICHs is likely higher than reported, as only a fraction of infants with an ICH present with clinical features [54]. Diagnostic imaging plays an important role in the detection of ICHs. A cranial US is a first-line modality for the evaluation of a neonate suspected of having an ICH. USs are a highly sensitive and dependable tool for assessing the ventricular system and central brain structures. It is highly effective in detecting germinal matrix hemorrhages, intraventricular hemorrhages, and hemorrhages in central brain structures [53]. However, a limitation of the US is its poor visualization of the peripheral and deep brain regions. Using additional acoustic windows, like the mastoid fontanelle, posterior fontanelle, or foramen magnum, can help overcome these limitations. In cases where more detailed information about ICH lesions is necessary, or if there are still suspicions despite there being normal cranial US results, MRI is the preferred imaging method. Unlike USs, MRI can capture images of the entire brain, as it is not limited by acoustic windows. Furthermore, MRI has the ability to utilize hemorrhage-sensitive sequences like susceptibility-weighted imaging (SWI), which is unparalleled in its sensitivity in detecting hemorrhages [57]. Therefore, MRI is considered the gold standard for further imaging when more comprehensive information is needed. However, there are also limitations to the use of MRI. In the neonatal period, MRI typically requires general anesthesia. Additionally, patients with a suspected ICH are often located in the neonatal intensive care unit, and they may be difficult to move to an MRI machine in another department; therefore, MRI is not always feasible. A CEUS is an emerging radiological modality that could prove especially useful in the detection of ICHs. The CEUS is a technique based on the vascularity and perfusion of the observed organ. As such, it may be an ideal tool to detect areas without perfusion, such as hemorrhages. The areas with normal perfusion greatly differ in terms of signal from that of the areas without perfusion, which should also improve the visualization of such areas in the peripheral and deep regions of the brain. Several recent studies have already described cases of ICHs that were more accurately diagnosed using CEUSs in comparison to that using MRI [20,51]. CEUS shows preserved cerebral perfusion around the hemorrhage (i.e., the presence of microbubbles within the brain) and the region of heterogeneous hypoperfusion in the affected area of the brain with the hypoechoic hemorrhagic core (i.e., no microbubbles entering the hematoma) (Figure 3).

### 3.4. Brain Tumors

Brain tumors are the most common solid tumors in the pediatric population and are the second most frequent type of childhood cancer overall. While these tumors can occur at any age, they are most prevalent in children aged 3–7 years. Tumors in neonates and infants up to the age of 2 years typically occur in the supratentorial region, while in children older than 2 years, they are more frequently found in the infratentorial region. There are more than a hundred different histological subtypes of brain tumors that have been recognized; the most common ones are pilocytic astrocytomas, brainstem gliomas, and medulloblastomas [58]. Imaging plays a vital role in the management of patients with brain tumors. The gold standard for brain tumor assessments is MRI. Advancements in neuroimaging MRI techniques, such as the use of diffusion-weighted imaging (DWI), spectroscopy, perfusion imaging, and functional MRI, provide additional information about the metabolism and physiology of these tumors, which can aid in their diagnosis and monitoring [59]. The value of CEUS in the management of brain tumors is expanding, but the information that is available so far is still limited. CEUSs can help in the detection of solid brain tumors because they are typically more vascularized than brain parenchymas are (Figure 4). However, its ability to differentiate between different subtypes is limited [60,61]. So far, it has been most extensively used in intraoperative settings for surgical guidance [62,63]. Neoplastic tissue shows a higher contrast enhancement compared to that of the normal surrounding brain parenchyma because of its higher vessel density. Thus, CEUSs allow the most precise guidance of the surgical procedure and is especially useful for resection control [60,64,65,66].

### 3.5. Vascular Malformations and Brain Vasculature Evaluation

The ISSVA (International Society for the Study of Vascular Anomalies) classifies vascular malformations into simple and combined malformations. The former ones are more common and include arteriovenous malformations (AVMs), venous, lymphatic, and capillary malformations, and arteriovenous fistulas. Typical pediatric vascular malformations are the vein of Galen malformations (a type of AVM), malformations of the dural sinus, and juvenile spinal vascular malformations [67]. The vein of Galen malformations are rare intracranial anomalies; however, the mortality rate in the absence of a treatment is nearly 100% [68]. Transarterial embolization is the treatment of choice, with the surgical treatment typically having high morbidity and mortality rates [69]. A CEUS can be helpful in establishing the diagnosis of the malformation, as the dense vascular structure of the malformation is better depicted using CEUSs in comparison to conventional USs. Additionally, after a surgical procedure or neurovascular embolization, brain CEUSs can be used to assess the residual flow within the lesion [3]. This has been more extensively researched for the other types of AVMs in the adult population [70,71,72,73]. By using CEUSs, angiography, CT angiography, or MRI can be avoided. 

CEUSs can also be a great problem-solving tool in the assessment of brain vasculature, especially the evaluation of brain sinuses for the presence of thrombosis. Due to the slow flow in the brain sinuses, MRI can be unreliable in diagnosing sinus thrombosis [74]. CEUSs using an approach through the anterior fontanelle enables a good visualization of the sagittal sinus, while the transtemporal and transmastoid approaches provide a great visualization of the transverse sinuses (Figure 5).

### 3.6. Infections

Neuroinfections can arise from various organisms, most commonly bacteria, viruses, or fungi. Clinically and radiologically, central nervous system infections are categorized as meningitis, cerebritis, an abscess, ventriculitis, extra-axial collections, or combinations of these [75]. Meningitis is primarily clinically diagnosed and confirmed through CSF evaluation. The role of imaging is to identify contraindications for a lumbar puncture, monitor complications associated with meningitis, and identify any underlying causes for recurrent meningitis. For neonates and infants, a cranial US has a crucial role, while CT, and especially, MRI with contrast administration are used for the assessment of meningitis and its complications among older children [76,77]. Although a CUES should, in theory, be an ideal imaging method to demonstrate the thickened and hypervascular leptomeninges in infants with meningitis, we found no studies reporting the use of CEUSs on this group of children. 

When an infection spreads to the brain parenchyma, it leads to cerebritis, which can progress to abscess formation. Cerebritis can be focal, diffuse, or bilateral, and imaging can demonstrate the involvement of the grey matter, particularly using diffusion-weighted MR imaging and FLAIR sequences [75,78]. CEUSs, with their quantitative analysis of brain perfusions, have the potential to be used to diagnose this entity. Even more so, CEUSs can be used to diagnose brain abscesses and differentiate them from other focal brain lesions. Brain abscesses typically appear as a non-enhancing focal lesion with a hyper-enhancing rim (Figure 6) [79].

### 3.7. Brain Death Confirmation

Brain death refers to the complete and irreversible loss of brain function, which is defined as the cessation of cortical and brainstem activities. Typically, the confirmation of brain death involves verifying three criteria: unconsciousness, the absence of brainstem reflexes, and apnea. However, the apnea test can be challenging to implement for infants and may pose risks to patients with unstable circulation. Therefore, various ancillary imaging tests can assist in making the diagnosis. Cerebral angiography and radionuclide scanning are commonly used on infants as ancillary imaging tests. However, they can be challenging to perform when the patient needs to be transported from an intensive care unit [80]. The bedside imaging test to confirm cerebral circulatory arrest is Doppler ultrasonography, which is used to evaluate the blood flow in intracranial and extracranial arteries. However, it may face challenges with transmission due to the inadequate penetration of ultrasound beams through the temporal bone, making the evaluation of intracranial vessels unreliable in some cases [81]. CEUSs allow the better visualization of cerebral vasculature compared to that of Doppler USs, and they also facilitate the evaluation of cerebral perfusion (Figure 7) [82]. Studies on adult populations have shown that the rate of inconclusive Doppler US examinations for determining cerebral circulatory arrest significantly decreases when CEUSs are performed [81,83]. A couple of reports already demonstrated that the CEUS can also be reliably used as an ancillary imaging test for the confirmation of brain death in neonates [37,84].

## 4. Conclusions

A CEUS is an increasingly used diagnostic imaging technique. It has proven to be a valuable tool in the assessment of numerous central nervous system conditions among children and can complement the conventional US, CT, or MRI. Its high safety profile makes it especially applicable to the pediatric population. The current evidence suggests that the potential risks associated with brain CEUSs are minimal and lower compared to those of other contrast agents like iodinated and paramagnetic contrast techniques. The assessment and quantification of cerebral perfusion using a CEUS provides unique functional insights into the pathophysiology of the brain. The future use of brain CEUSs for established applications and research on the potential value of brain CEUSs for additional clinical applications are, thus, highly justified.

## Figures and Tables

**Figure 1 children-10-01303-f001:**
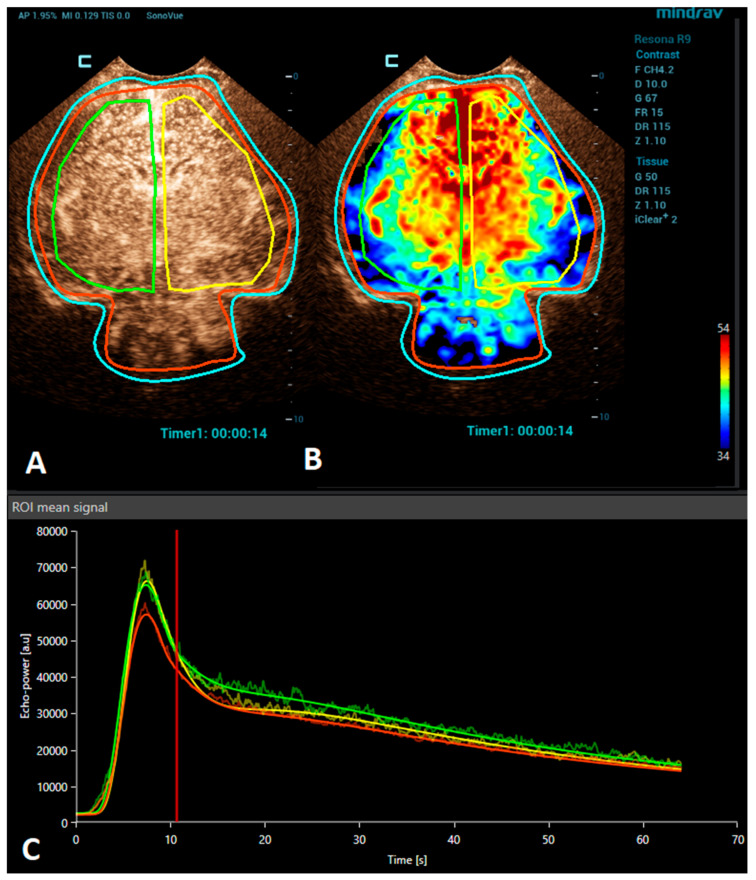
A VueBox^®^ analysis of a 4-day-old premature girl with neonatal sepsis and normal symmetric CEUS brain perfusion. Regions of interest were set for: the whole brain/both hemispheres (red), right hemisphere (green), and left hemisphere (yellow). (**A**) The above-left panel displays a coronal CEUS image, demonstrating symmetric brain perfusion during the arterial phase of enhancement. (**B**) The above-right panel shows a normal symmetric coronal perfusion map of the average contrast signal intensity (MeanLin) parameter. (**C**) The below panel presents time–intensity curves for the right hemisphere (green) and left hemisphere (yellow) and the whole brain (red). A subsequent brain MRI scan was normal.

**Figure 2 children-10-01303-f002:**
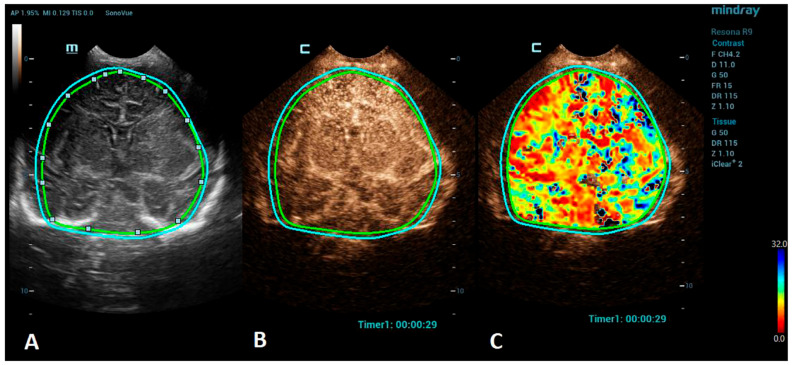
A VueBox^®^ analysis of a 5-day-old male with ischemic stroke in the territory of the left middle cerebral artery. (**A**) The coronal grayscale reference ultrasound image of the brain. (**B**) The coronal CEUS image. (**C**) The coronal CEUS perfusion map highlighting an increased fall time parameter in the territory of the left middle cerebral artery. The whole brain is selected as a region of interest in the presented image (marked by the two color lines).

**Figure 3 children-10-01303-f003:**
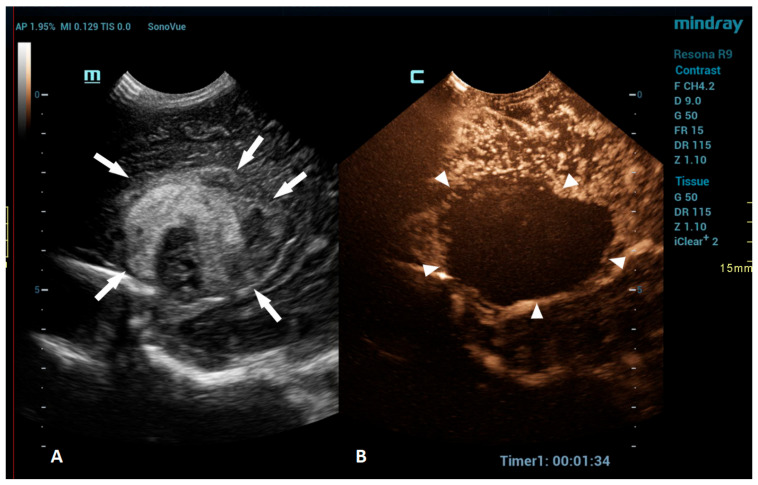
A 1-day-old boy who presented with apnea. (**A**) The sagittal grayscale ultrasound brain image reveals a heterogeneous lesion (arrows) in the right frontotemporal region. (**B**) The sagittal CEUS image confirms avascularity of the lesion (arrowheads), consistent with the hemorrhage, and no obvious vascular malformation.

**Figure 4 children-10-01303-f004:**
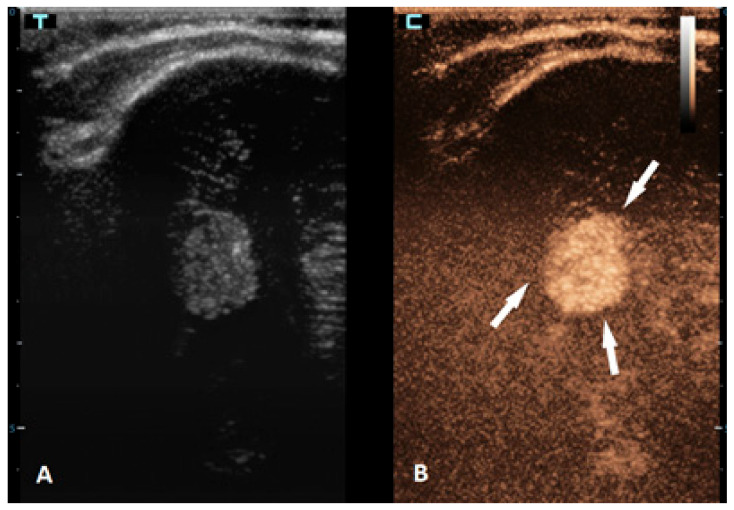
A 6-month-old premature boy with an accidentally detected small lesion in the 4th ventricle. (**A**) The transverse reference grayscale ultrasound image through the left mastoid fontanelle. (**B**) The transverse CEUS image of the 4th ventricle demonstrates avid arterial and venous enhancement of the lesion (arrows). Findings were suggestive of choroid plexus papilloma; subsequent brain MRI scans confirmed benign aetiology of the lesion.

**Figure 5 children-10-01303-f005:**
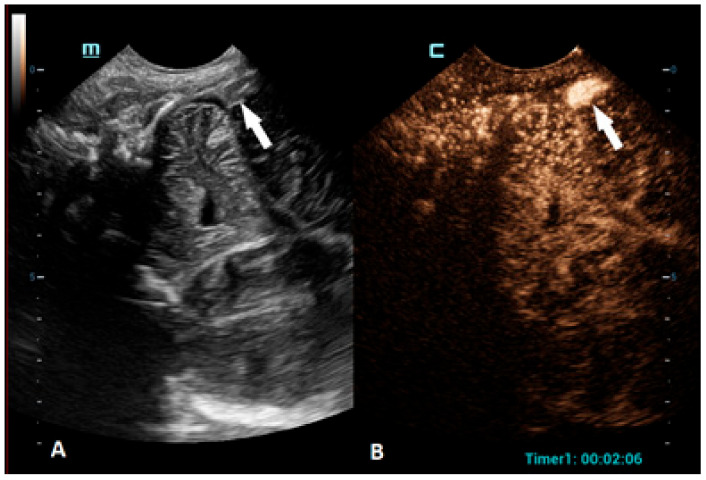
A 5-day-old boy diagnosed with hypoxic-ischemic encephalopathy was investigated for suspected thrombosis of the right transverse sinus (suspected on MRI scan). (**A**) The transverse reference grayscale ultrasound image through the right mastoid fontanelle and increased echogenicity within the transverse sinus (arrow). (**B**) The CEUS image of the right transverse sinus shows the sinus with high signal intensity (arrow), ruling out thrombosis.

**Figure 6 children-10-01303-f006:**
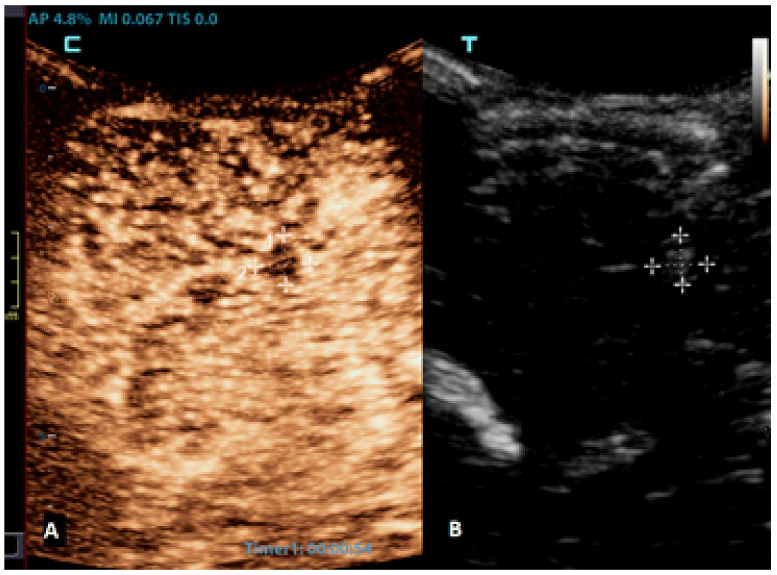
A 2-month-old boy with acute lymphoblastic leukaemia and candida sepsis. Slowly growing hyperechoic lesions were detected using a brain ultrasound. (**A**) The sagittal CEUS image of the brain at 54 s after the application of contrast agent shows high signal intensity only at the rim of the lesion (crosses). Candida micro-abscesses were suspected and later confirmed using a brain MRI scan. (**B**) A sagittal reference grayscale ultrasound image shows a small hyperechoic lesion (plusses).

**Figure 7 children-10-01303-f007:**
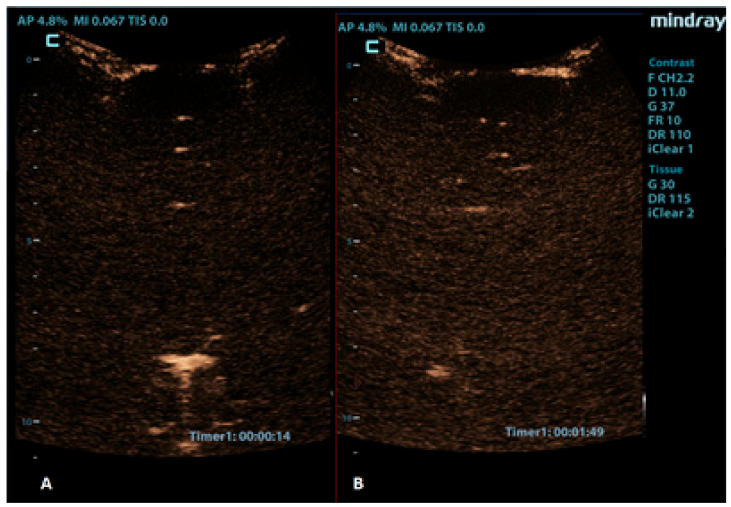
Coronal CEUS images of a 1-year-old infant’s brain obtained (**A**) 14 s and (**B**) 109 s after the contrast administration. Both images demonstrate a lack of enhancement of the intracranial vasculature and no brain perfusion.

## Data Availability

Not applicable.

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
