# Peer review of "The Use of Contrast-Enhanced Ultrasound (CEUS) in the Evaluation of the Neonatal Brain"

_children, 2023, doi:10.3390/children10081303_

Round 1

Reviewer 1 Report

Thanks for the opportunity to review this manuscript on CEUS in evaluation of neonatal brain. It is important to have regular US prior to a CEUS. This has not been mentioned in the manuscript. I felt the manuscript is similar to a few other reviews on the topic.

Author Response

We would like to thank the reviewer for the insightful comments and valuable recommendation. We completely agree that performing a grayscale US (i.e. B-mode US) is important prior to performing CEUS. Following the reviewer's recommendation we have added a couple sentences to the beginning of paragraph 2.2 (Contrast-enhanced ultrasound (CEUS) technique) to emphasize that. The start of this paragraph now reads as follows:

"Before performing CEUS, it is important to perform a detailed grayscale US examination to evaluate the region of concern. This initial assessment can be beneficial in customizing the subsequent CEUS protocol."

Reviewer 2 Report

Only a few clarifications are necessary

Abstract: the first paragraph seems rather useless to me, it says widely known things, I recommend reducing or modifying it

point 2.2 and 2.3 have the same name: please verify the correct name and modified accordigly

Author Response

We would like to thank the reviewer for these insightful comments and advice on how to make our article stronger.

Comment 1:
Abstract: the first paragraph seems rather useless to me, it says widely known things, I recommend reducing or modifying it

Response1:
Following the reviewer's recommendation, we have modified the abstract and the first paragraph of the manuscript to reduce the amount of the widely known information.

Comment 2:
point 2.2 and 2.3 have the same name: please verify the correct name and modified accordigly

Response2:
We want to sincerely thank the reviewer for this keen observation and we apologize for the mistake. The point 2.3 had the wrong subtitle. We have corrected this mistake in the revised version of the manuscript. The correct subheading of point 2.3 is "Quantification methods".

Reviewer 3 Report

The authors present a very interesting and comprehensive review of the issue of CEUS for the examination of the brain in neonates.

I consider a positive aspect that, in addition to the specific application, the authors also approach the general and physical principles of CEUS functioning as well as the implementation of a practical approach (how to perform it) - considering the situation where, in the case of the European Union, CEUS is still an off-label indication for pediatric patients, for doctors primarily engaged in imaging in pediatric patients, the theoretical overview of CEUS functioning may not be generally known.

I appreciate the separate section devoted to safety and the current situation in terms of the approval of contrast agents for use in children in case of FDA and EMA.

The authors discuss specific applications of CEUS in specific clinical situations and supplement these with a valuable review of the literature and available studies. I also appreciate the fact that most of the listed applications are illustrated by the authors with images from their own clinical practice.

The article serves as an excellent overview for the reader who wants to get an overview of the issue, but it also represents a reasonable basis for potential research in practice.

I caught one mistake - on line 188 there is the word "hyperthermia" instead of "hypothermia".

I have no major objections to the article.

Author Response

We truly appreciate the reviewer’s comments and sincerely thank the reviewer for emphasizing the strong points of our manuscript. The current literature with practical advice and images of brain CEUS is scarce and we hope our article with practical information helps to promote the use of this valuable method. 

Thank you for the comment regarding "hyperthermia" (line 188) - we sincerely apologize for this mistake. The mistake has been corrected in the revised version of the manuscript and now reads "hypothermia".

Round 2

Reviewer 1 Report

Thanks for the revisions. The manuscript reads well.